# Coupled and Coordinated Development of the Tourism Industry and Urbanization in Marginal and Less Developed Regions—Taking the Mountainous Border Areas of Western Yunnan as a Case Study

**Pengyang Zhang [1,2,*], Lewen Zhang [1], Dandan Han [1], Tingting Wang [1], He Zhu [3] and Yongtao Chen [4]**

1 School of Business and Tourism Management, Yunnan University, Kunming 650500, China
2 Research Center for Border Tourism, China Tourism Academy, Kunming 650500, China
3 Institute of Geographic Sciences and Natural Resources Research, Chinese Academy of Sciences, Beijing 100101, China
4 Lancang-Mekong International Vocational College, Yunnan Minzu University, Kunming 650500, China
* Correspondence: young@ynu.edu.cn

**Abstract:** Promoting the coupled and coordinated development of China's tourism industry and urbanization is of great significance for its marginal and less developed regions. Taking a typical marginal and less developed region, the mountainous border areas of Western Yunnan as the research object, this study analyzed the spatial and temporal characteristics of the coupling coordination degree of the tourism industry and urbanization, as well as their influencing factors, in this region from 2010 to 2019 using the coupling coordination model, spatial gravity model, and panel Tobit model. The study results show the following. (1) The development level of the tourism industry and urbanization in the study region had significantly increased, but there was an obvious polarization phenomenon in its spatial distribution. (2) The coupling coordination degree of the tourism industry and urbanization showed a good development trend of steady growth, and the areas were ranked according to the average annual growth rate as follows: West Yunnan > Southwest Yunnan > Northwest Yunnan. (3) The regional differences in the coupling coordination degree had expanded, reflecting an "agglomeration phenomenon" and "distance decay effect", and the tourism industry lagging (obstructed) subtype was dominant. (4) The industrial structure, transportation accessibility, capital effect, consumption capacity, and talent support had significant positive effects on the coupling coordination degree, but the role of openness to the outside world was not obvious. This study can provide a useful reference for further studies on the marginal and less developed regions of China.

**Keywords:** tourism industry; urbanization; coupling coordination; panel Tobit model; the mountainous border areas of western Yunnan





## 1. Introduction

Many countries around the world face the problem of uneven urbanization [1], especially developing countries [2]; these countries' uneven urbanization is mainly reflected in the fact that the development of some of their less developed regions tends to be hampered by various development difficulties [3]. Tourism is a less demanding economic sector than other industries, but as a comprehensive industry, it can have a significant impact on regional development [4–6]. Many studies have confirmed the positive effects of tourism on the alleviation of regional imbalances [7–9], and its improvement of urbanization imbalances is one of the industry's most important aspects [10]. From a "core–periphery" perspective of a country's regional development levels, it can be seen that some of its less developed regions are generally located in the interior periphery of the country [3,11] and are characterized by a negative migration balance, low living standards, ageing populations, few employment opportunities, low levels of education, low regional investment [12],

and difficulties in implementing the traditional urbanization model, which has disadvantages, such as "adventurous urbanization", "pseudo-urbanization", "urban disease", and "urban–rural imbalance" [13]. In contrast, urbanization in the new era is people-oriented and quality-oriented, and it aims to integrate urban and rural and green development [14]. In this era, tourism is seen as an effective way to promote economic and local development, especially in marginal areas, such as mountainous regions [15,16]; therefore, using tourism to promote urbanization [17] is one of the best options for the development of marginal and less developed areas [18,19]. However, due to the particular realities of marginal and less developed regions [20], the inherent relationship between tourism and urbanization in these regions may exhibit different characteristics to other regions. It is important, therefore, to examine the relationship between the tourism industry and urbanization, as well as to identify the interactive evolutionary process of the two in order to guide the development of marginal and less developed regions.

In 1991, Mullins first proposed the concept of "tourism urbanization", which is a kind of urbanization based on consumption that is different from traditional industrial urbanization [21]; after this, the relationship between tourism and urbanization began to attract academic attention, and the characteristics [22], driving mechanisms [23], types [24], models [25], and impacts [26] of tourism urbanization have been richly discussed. The relationship between tourism and urbanization can be summarized into three main types: first, tourism promotes urbanization [27], accelerates the expansion of towns and cities through tourism development [28], leads to population concentration [29], changes the regional industrial structure [30], and becomes a new engine of urbanization [31]; second, urbanization that promotes tourism development [32], urbanization based on the combined effect of consumption and investment, and driving the expansion of the scale of the tourism industry [33] will all promote increases in tourism output [34]; third, tourism and urbanization are considered to have a two-way interaction [35], with tourism development and urbanization being highly interactive and dependent [36]. Mass tourism has been found to cause an increase in land consumption, driving urbanization, and urban expansion has consolidated tourism development in subsequent phases [37]. However, in terms of research methods, the relationship between the tourism industry and urbanization has mostly been evaluated by using causality tests [38], regression analysis [39], coupling coordination models [40], and other methods; there are few studies on the mechanism construction, spatial–temporal evolution, and influencing factors of the interaction between the two. In terms of research scales, studies have covered the national [41], regional [42], provincial [43], and city [44] levels, but not enough attention has been paid to marginal, less developed areas.

Some marginal and less developed regions in China are rich in tourism resources, but they lag behind in their urbanization construction. Therefore, in the context of urbanization construction, how to effectively articulate the relationship between the tourism industry and urbanization and how to more effectively promote the synergistic development of the two are of great significance for promoting the development of marginal and less developed regions. In view of this, based on the analysis of the mechanism of coupling and coordination between the tourism industry and urbanization, we selected a typical marginal and underdeveloped region, the mountainous border areas of western Yunnan, as the research object, and analyzed the coupling and coordination between the tourism industry and urbanization in this region, as well as the spatial–temporal evolution characteristics and influencing factors of the interaction between the two using the panel Tobit model. This study mainly addressed the following questions. 1. What is the mechanism of the coupling and coordination between the tourism industry and urbanization? 2. How large is the tourism industry and the extent of urbanization in the mountainous border areas of western Yunnan? 3. What is the spatial–temporal evolution of the degree of coupling coordination between the tourism industry and urbanization? 4. What are the spatial combination and evolution patterns of different types of coupling coordination? 5. Which factors affect the

coupling coordination degree? The study's main conclusions and recommendations will be of reference value for the development of marginal and less developed regions.

## 2. The Mechanism of Coupling and Coordination between the Tourism Industry and Urbanization

Modern tourism development has begun to turn "people-oriented", and urbanization is also "people-oriented", emphasizing the value of urbanization from perspectives ranging from structuralism to humanism [45]. Thus, the common goal of humanism provides the value basis for the coupling and coordination of the tourism industry and urbanization. On the one hand, tourism, as an intrinsic driving force for industrial transformation, social change, and economic growth, can effectively contribute to the restructuring of urban space, the transformation of urban functions, and the improvement of urban quality [46]. On the other hand, accelerating the construction of urbanization with people as its core can more effectively release the promotion effect of urbanization on tourism economic development and provide adequate support for tourism industry development [47]. Based on this, we constructed an analysis framework of the coupling and coordination mechanism of the tourism industry and urbanization, which is based on the main line of the interaction between the two (see Figure 1).

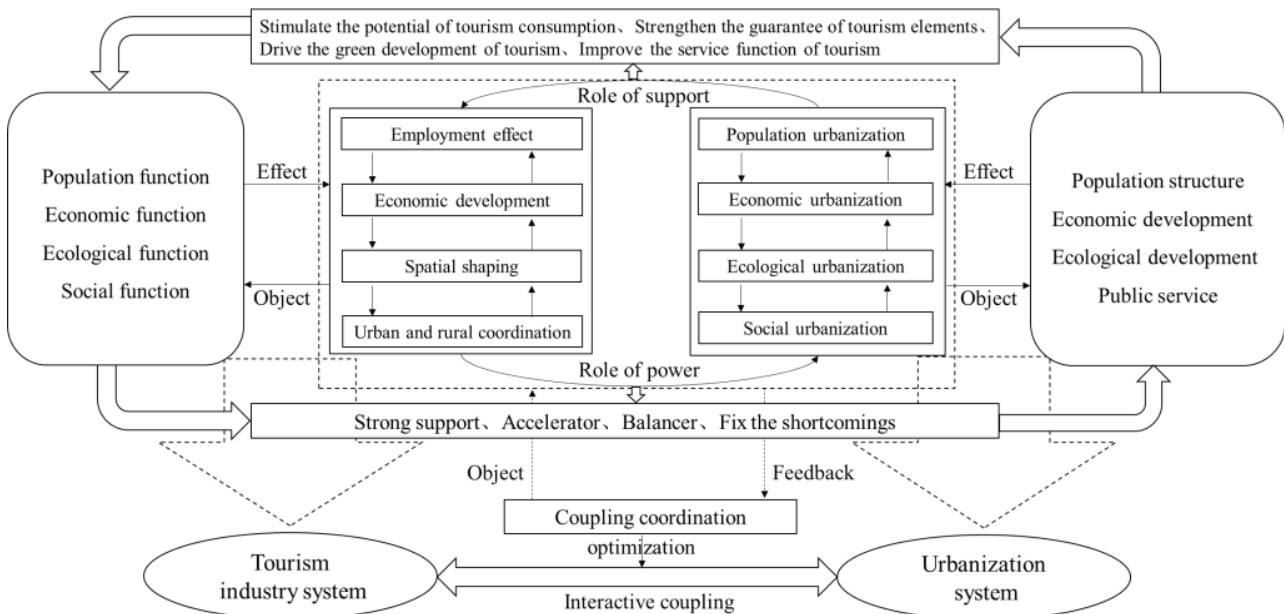

**Figure 1.** The coupling and coordination mechanism of tourism industry and urbanization.

### 2.1. Tourism as a Key Driving Force of Urbanization

The tourism industry as a driving force of urbanization is mainly manifested in four aspects. (1) The tourism industry helps create employment opportunities and provides "strong support" for urbanization. Compared with other industries, tourism employment has a large capacity, low threshold, and is inclusive and flexible, along with other characteristics, which are conducive to solving the problem of total employment expansion and structural optimization in the process of urbanization. Tourism development leads to great employment gaps, which help to promote the non-agricultural engagement of the rural migrant labor force [48] and therefore the transfer of populations to urban areas. In particular, tourism employment or entrepreneurship in underdeveloped areas is also conducive to the local and nearby employment of farmers [49], reducing the "migratory bird" migration problem that affects a large number of migrant workers under the urban–rural dual system and providing strong support for population urbanization. (2) The tourism industry can help drive economic growth [50] and provide a "gas pedal" for urbanization. The tourism industry has the characteristics of strong comprehensiveness, high correlation, being a

strong driving force, etc. Through the tourism industry's agglomeration and integration, it can directly or indirectly drive the economic growth of a given town. The tourism industry agglomeration process promotes the convergence of urbanization factors, such as human flow, logistics, capital flow, etc. Furthermore, it can accelerate the industrial structure adjustment of a given town. Tourism industry integration refers to the "tourism +" or "+ tourism" method of achieving the integration and interactive development of the tourism industry and multiple industries, as well as of fully releasing the "multiplier effect" of tourism in the construction of urbanization. (3) The tourism industry helps shape environmental space and provides a "balancer" for urbanization. A new concept of tourism development, all-for-one tourism emphasizes a transition from "point-line type" to a more open "plate type" tourism space system focusing on scenic spots [51] to a certain extent, which can eliminate the regional differences caused by non-homogeneous tourism resource backgrounds so as to promote tourism-driven urbanization on a larger scale. At the same time, the "green" attributes of the tourism industry are highly compatible with environmental protection requirements in the construction of urbanization. Firstly, tourism development can ease the environmental pressure brought by urban expansion. Secondly, participation in tourism activities can subconsciously enhance the environmental awareness of tourists. (4) The tourism industry helps balance urban and rural development and "make up for weakness" of urbanization construction. With the rapid development of rural tourism, a large number of tourism flows converge in the countryside, which can improve the income of local rural residents, promote local infrastructure construction, and guide local development [52]. In addition, the market supply of town tourism cannot be separated from rural support. Town tourism consumption also affects the rural economic system through various transmission mechanisms and dynamic effects. Therefore, as a bridge between town and rural synergistic development, the tourism industry can effectively alleviate the dual structure difficulties involved in the construction of urbanization.

### 2.2. Urbanization as Efficient Guarantee Support for Tourism Industry

The supporting role of urbanization for the tourism industry is mainly manifested in four aspects. (1) With population urbanization as its core, urbanization can stimulate tourism consumption potential. Urbanization not only focuses on increasing the proportion of the non-farm population, but it also pays attention to the "citizenship" status of the population after its convergence in cities and towns [53], which drives tourism consumption in terms of quantity and quality. On the one hand, a large number of people continue to integrate into cities and towns, and their emphasis on leisure and relaxation has increased, leading to an increase in demand for tourism consumption. On the other hand, the overall income level of the residents involved in the urbanization process has increased, and the demand for tourism consumption has become more diversified, thus accelerating the structural reform of the supply side of the tourism industry. (2) Urbanization focuses on economic urbanization, which can strengthen the guarantee of tourism elements. Industrial agglomeration is an essential carrier of urbanization, and the factor agglomeration effect brought about by urbanization also guarantees industrial agglomeration [54]. During urbanization construction, investment and financing channels are expanded, which can provide more financial security for tourism development. Large numbers of people being concentrated in cities and towns can also provide sufficient labor sources for tourism. In terms of technological production factors, the internet, big data, and other information technologies continue to promote the construction of urbanization, and future technological empowerment will further promote the transformation and upgrading of tourism. (3) Urbanization involves ecological urbanization, which can drive the green development of tourism. Urbanization emphasizes the concept of an ecological civilization and pays attention to environmental protection in urban construction [55]. Especially under the guidance of the scientific development concept, ecological protection and resource conservation are essential. Future urban construction will restrict high-polluting and high-consumption industries and support environmentally friendly modern industries,

such as tourism. Furthermore, the mode of development adopted for urbanization should be efficient and based on scientific evidence. It should reduce the unnecessary waste of resources caused by inefficiency, significantly contributing to the improvement of the tourism industry's business environment and helping the industry to further exploit the advantages of green development. (4) Urbanization is based on social development, which can improve tourism service functions. The continuous improvement of infrastructure and public services is an essential requirement in the development process of urbanization [56]. The construction of urban facilities, such as accommodation, entertainment, and shopping facilities, guarantees the realization of the tourism elements' functions, and the tourism service functions that rely on urban construction also tend to be optimized, leading to, for instance, more convenient tourism transportation services, increases in tourism information levels, and the continuous improvement of tourism market governance, which is conducive to the creation of more competitive tourism destinations.

## 3. Research Design

### 3.1. Research Area

The mountainous border areas of western Yunnan (Figure 2), China, are located in the southern part of the Hengduan Mountains and the intermountain basin of southern Yunnan. The region includes Baoshan, Lijiang, Pu'er, Lincang, Chuxiong Yi Autonomous Prefecture, Honghe Hani and Yi Autonomous Prefecture, Xishuangbanna Dai Autonomous Prefecture, Dali Bai Autonomous Prefecture, Dehong Dai Jingpo Autonomous Prefecture, and Nujiang Lisu Autonomous Prefecture. Our main reasons for selecting the mountainous border areas of western Yunnan as the study area were as follows. (1) Regional typicality. The mountainous border areas of western Yunnan are not only the border areas of western China, but are also contiguous poverty-stricken areas and ecologically fragile areas. They are typical representatives of marginal and less developed regions. (2) Research importance. In 2021, the urbanization rate of the mountainous border areas of western Yunnan was only 47.29%, which was 3.76 percentage points lower than the urbanization rate of Yunnan Province (51.05%) and 17.43 percentage points lower than the urbanization rate of China as a whole (64.72%). The mountainous border areas of western Yunnan account for more than half of Yunnan Province, and the urbanization development of this area is related to the overall promotion of Yunnan's urbanization construction. (3) Urgent need for attention. The mountainous border areas of western Yunnan are rich in tourism resources and have a sound foundation for the development of the industry in these areas. Therefore, with relevant, favorable policy, the mountainous border areas of western Yunnan could see significant tourism development. In order to realize these areas' great-leap-forward development, it is necessary to explore a path of the synergistic development of the tourism industry and urbanization.

### 3.2. Construction of Index System and Data Sources

3.2.1. Comprehensive Evaluation Index System of Tourism Industry

Two evaluation methods are used to assess tourism industry development levels: the single index evaluation method and the comprehensive index evaluation method. The single index method primarily uses total tourism revenue or total tourist arrivals. However, it cannot accurately measure the tourism industry's development performance, and it cannot easily be used to analyze the internal mechanism involved in the interaction between tourism and urbanization. In this paper, based on the existing research, and following the principles of the representativeness, independence, and accessibility of indicator selection, nine secondary indicators were selected from four dimensions: economic benefits, market scale, industrial scale, and development potential (see Table 1).

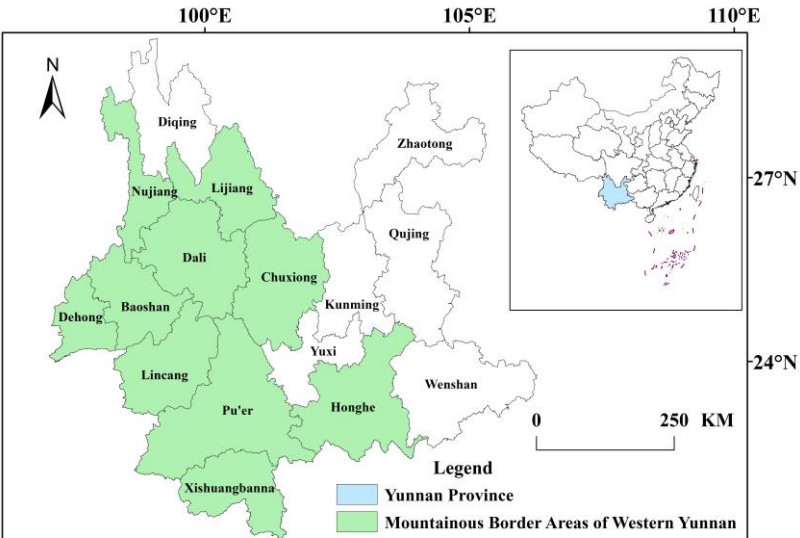

**Figure 2.** Research area. Note: This map is based on the standard map of GS (2019)1822, downloaded from the standard map service website of the Ministry of Natural Resources. The base map is not modified, and the same applies below.

### 3.2.2. Comprehensive Evaluation Index System of Urbanization

The evaluation indexes of urbanization levels used in academia mainly involve three dimensions: population, space, and economy. However, some scholars have expanded these indexes to include social services, ecological environment, and lifestyle dimensions [57–60], with these expanded indexes providing a more scientific understanding of the connotations of urbanization. At present, there is scholarly consensus that China's urbanization road has shifted from "land urbanization" to "human urbanization" [61], so it is necessary to establish development-oriented evaluation index systems. In this paper, based on the analysis of the coupling coordination mechanism, and following the principles of the representativeness, independence and accessibility of indicator selection, 20 secondary indicators were selected from four dimensions: population structure, economic development, ecological environment, and public services (see Table 1).

### 3.2.3. Data Sources

This study examined the interval from 2010 to 2019, and its target region was the mountainous border areas of western Yunnan, with a sample area of 10 cities in Yunnan Province used, namely, Baoshan, Lijiang, Pu'er, Lincang, Chuxiong, Honghe, Xishuangbanna, Dali, Dehong, and Nujiang. The original data were obtained from the Yunnan Statistical Yearbook (2011–2020), the statistical yearbooks of the cities in the mountainous border areas of western Yunnan, and statistical bulletins on socio-economic development and official government websites, and the missing data of some indicators were processed by linear simulation and collated by calculation.

### 3.3. Empirical Method

The use of quantitative methods to study the relationship between the tourism industry and urbanization can demonstrate the intrinsic link between them more intuitively. Although qualitative methods can show that the tourism industry and urbanization influence each other, there are limitations in the in-depth analysis of the degree of their mutual influence, the evolution of their relationship, and the influencing factors in their relationship. Therefore, this study integrated the entropy method, linear weighting method, coupling coordination model, space gravity model, and panel Tobit model to study the relationship between the tourism industry and urbanization.

**Table 1.** The comprehensive evaluation index system and weight of the tourism industry and urbanization of the mountainous border areas of Western Yunnan.

| Goal Layer | Criterion Layer | Indicator Layer | Unit | Index Attribute * | Weights |
|---|---|---|---|---|---|
| Tourism industry | Economic benefit | Total tourism income | Billion CNY | + | 0.155 |
| | | Tourism foreign exchange earnings | Billion USD | + | 0.196 |
| | Market size | Domestic tourist arrivals | Million people | + | 0.105 |
| | | Overseas tourist arrivals | Million people | + | 0.178 |
| | Industry scale | Number of travel agencies | Ind | + | 0.161 |
| | | Number of scenic spots | Ind | + | 0.071 |
| | | Number of employees in the accommodation and catering industry | Million people | + | 0.086 |
| | Development potential | The growth rate of domestic tourist arrivals | % | + | 0.013 |
| | | The growth rate of overseas tourist arrivals | % | + | 0.035 |
| Urbanization | Population structure | Proportion of urban population | % | + | 0.026 |
| | | Population density | People/square kilometer | - | 0.057 |
| | Economic development | Number of industrial enterprises above designated size | Ind | + | 0.066 |
| | | Number of key service enterprises | Ind | + | 0.047 |
| | | Non-agricultural industry as a proportion of GDP | % | + | 0.039 |
| | | Disposable income of urban residents | CNY | + | 0.037 |
| | | Per capita GDP | CNY | + | 0.046 |
| | Ecological environment | Comprehensive index of air pollution | mg/m$^3$ | - | 0.037 |
| | | Green coverage rate of built district | % | + | 0.016 |
| | | Number of days when the air quality reaches and is better than the second level | Per day | + | 0.016 |
| | | Treatment rate of domestic sewage | % | + | 0.021 |
| | | Per capita public green space area | Square meters | + | 0.028 |
| | Public services | Postal traffic business volume as a proportion of GDP | % | + | 0.049 |
| | | Public libraries | Ind | + | 0.073 |
| | | Number of beds in health institutions | Sheet | + | 0.070 |
| | | Passenger car ownership by city | 10,000 trucks | + | 0.073 |
| | | General public budget expenditure | Billion CNY | + | 0.054 |
| | | Extent of graded roads | Kilometers | + | 0.052 |
| | | Number of villages benefitting from tap water | Ind | + | 0.081 |
| | | Rural electricity consumption | Billion kilowatt hours | + | 0.112 |

\* Note: "+" represents a positive effect and "-" represents a negative effect.

### 3.3.1. Index Pretreatment and Weight Solving

In order to eliminate the influence of the number and scale differences of the indicators in the evaluation system of the tourism industry and urbanization on the calculation results, it was necessary to standardize the indicators to reduce the interference of random factors. Among them, the relevant indicators were divided into positive effect indicators and negative effect indicators, with different treatment formulae being used for these two types of indicators.

$$\text{Positive indicator}: Y_{ij} = \frac{\left(x_{ij} - x_{ij\min}\right)}{\left(x_{ij\max} - x_{ij\min}\right)} \tag{1}$$

$$\text{Negative indicator}: Y_{ij} = \frac{\left(x_{ij\max} - x_{ij}\right)}{\left(x_{ij\max} - x_{ij\min}\right)} \tag{2}$$

In the formulae, the following are defined: $Y_{ij}$ denotes the standardized value of the $j$th indicator in year $i$; $x_{ij}$ denotes the original value of the $j$th indicator in year $i$; $x_{ij\max}$ denotes the maximum value of indicator $j$; $x_{ij\min}$ denotes the minimum value of indicator $j$; $i = 1, 2, 3, \ldots, m$ denotes the number of years; and $j = 1, 2, 3, \ldots, n$ denotes the number of indicators.

The indicator weights reflect the indicators' relative importance and affect the evaluation results' reliability and accuracy. The entropy method, an objective assignment method, was used in this paper, which determines the weights according to the magnitude of the

variability of the indicators, thus avoiding the subjectivity of artificial assignment. The relevant formula is shown in the literature of Xu (2019) [62], and the specific weighting results for the indicators are shown in Table 1.

### 3.3.2. Comprehensive Evaluation Value of Tourism Industry and Urbanization

Based on the weights calculated by the entropy method combined with the data after standardization of indicators, using the system index assessment model, the linear weighting method was applied to calculate the subsystem evaluation values of the economic benefits, market scale, industrial scale, and development potential of the tourism industry, as well as the subsystem evaluation values of the population structure, economic development, ecological environment, and public services of urbanization, in order to arrive at the total evaluation value of the tourism industry and urbanization, which is calculated by the following formula:

$$f(x) = \sum_1^n w_i \times x_i, g(y) = \sum_1^m w_j \times y_i \tag{3}$$

$$F(x) = \sum_1^n W_i \times f(x), G(y) = \sum_1^m W_j \times g(y) \tag{4}$$

In the formulae, $f(x)$ and $g(y)$ denote the comprehensive evaluation value of the tourism industry subsystem and urbanization subsystem, respectively; $F(x)$ and $G(y)$ denote the comprehensive evaluation value of the tourism industry system and urbanization system, respectively; $x_i$ and $y_i$ denote the standardized values of the tourism industry evaluation index and urbanization evaluation index, respectively; $w_i$ and $w_j$ denote the weight of the tourism industry evaluation index and urbanization evaluation index, respectively; and $W_i$ and $W_j$ denote the weights of the tourism industry subsystem and urbanization subsystem, respectively.

### 3.3.3. Coupling Coordination Model

The coupling coordination model is borrowed from the capacity coupling coefficient model in physics to measure the coupling process for two or more systems that interact and influence each other, and its function is to calculate the degrees of coupling and coupling coordination between systems. The coupling degree refers to a given system's strength or the degree to which the system's internal elements act on each other. In contrast, the coupling coordination degree refers to the extent to which this coupling is good or bad. We drew on existing research [63] to construct a coupling coordination model for analyzing the characteristics of the coupling coordination relationship between the tourism industry and urbanization in the mountainous border areas of western Yunnan. The formulae are as follows:

$$C = 2 \times \left[ \frac{F(x) \times G(y)}{(F(x) + G(y))^2} \right]^{\frac{1}{2}} \tag{5}$$

$$D = \sqrt{C \times T}, T = \alpha F(x) + \beta G(y) \tag{6}$$

In the formula, $C$ is the coupling degree and $D$ is the coupling coordination degree; $F(x)$ is the comprehensive evaluation level of the tourism industry and $G(y)$ is the comprehensive evaluation level of urbanization; $T$ denotes the coupling coordination development level index; $\alpha$ and $\beta$ denote the specific weights of the tourism industry system and urbanization system, respectively, and mainly measure the importance of each system. As we believed that both systems were very rich in connotations, and as the contributions of the two systems were not distinguished from each other, we took $\alpha = \beta = 0.5$ here; the values of $C$ and $D$ were in the range of [0, 1]. In order to make a further objective evaluation of the coupling and coordination level of the two systems, we drew on the existing research results [64,65]. We divided the coupling degree type (Table 2) and the coupling coordination degree type (Table 3) into intervals.

**Table 2.** The classification of coupling degree.

| Coupling Degree Value | Coupling Level | Characteristic |
|---|---|---|
| $0.0 < C \leq 0.3$ | Severe uncoupling | Poor interconnectivity |
| $0.3 < C \leq 0.5$ | Slight uncoupling | Increased interconnectedness with simultaneous inhibition |
| $0.5 < C \leq 0.8$ | Primary coupling | System enters benign coupling |
| $0.8 < C \leq 1.0$ | Advanced coupling | Good resonance, jointly promote development |

**Table 3.** The classification of coupling coordination degree.

| Type | Numerical Value | Subtype | Numerical Value | Coupling Coordinator Subtype |
|---|---|---|---|---|
| Coordinated development | $0.8 < D \leq 1.0$ | Good coordination | $G(y) - F(x) > 0.1$ | Good coordination—Tourism industry lagging behind |
| | | | $F(x) - G(y) > 0.1$ | Good coordination—Urbanization lagging behind |
| | | | $0 < |G(y) - F(x)| \leq 0.1$ | Good coordination |
| | $0.6 < D \leq 0.8$ | Moderate coordination | $G(y) - F(x) > 0.1$ | Moderate coordination—Tourism industry lagging behind |
| | | | $F(x) - G(y) > 0.1$ | Moderate coordination—Urbanization lagging behind |
| | | | $0 < |G(y) - F(x)| \leq 0.1$ | Moderate coordination |
| Transformational development | $0.4 < D \leq 0.6$ | Minimal coordination | $G(y) - F(x) > 0.1$ | Minimal coordination—Tourism industry lagging behind |
| | | | $F(x) - G(y) > 0.1$ | Minimal coordination—Urbanization lagging behind |
| | | | $0 < |G(y) - F(x)| \leq 0.1$ | Minimal coordination |
| Uncoordinated development | $0.2 < D \leq 0.4$ | Close to imbalance | $G(y) - F(x) > 0.1$ | Close to imbalance—Tourism industry lagging behind |
| | | | $F(x) - G(y) > 0.1$ | Close to imbalance—Urbanization lagging behind |
| | | | $0 < |G(y) - F(x)| \leq 0.1$ | Close to imbalance |
| | $0.0 < D \leq 0.2$ | Severe imbalance | $G(y) - F(x) > 0.1$ | Severe imbalance—Tourism industry lagging behind |
| | | | $F(x) - G(y) > 0.1$ | Severe imbalance—Urbanization lagging behind |
| | | | $0 < |G(y) - F(x)| \leq 0.1$ | Severe imbalance |

### 3.3.4. Space Gravity Model

The space gravity model is widely used to analyze spatial linkages. It is a model for measuring the strength of spatial interactions between two regions based on the distance decay principle. To further characterize the spatial linkage pattern of the coupling coordination degree of the tourism industry and urbanization among the cities in the study region, drawing on existing studies [66], we constructed a spatial gravitational model as follows:

$$R_{ij} = K \times \frac{D_i \times D_j}{d_{ij}^b} \tag{7}$$

In Equation (7), $R_{ij}$ indicates the spatial linkage strength of the coupling coordination degree between two cities, $i$ and $j$; $D_i$ and $D_j$ indicate the coupling coordination degree $D$ value of city $i$ and $j$, respectively; $d_{ij}$ indicates the geographical distance between the two cities; $b$ is the distance–decay parameter, which took the value of 2 in this paper; and $K$ is the gravitational parameter, which took the value of 1 in this paper.

### 3.3.5. Panel Tobit Model

The Tobit model, also known as the sample model, as well as the restricted dependent variable model, were first proposed by Tobin in 1958 and are models in which the dependent variable takes on values that are continuous but subject to certain restrictions. In this study, considering that the measured coupling coordination degree of the tourism industry and urbanization had a value range of [0, 1] and belonged to the imputed data, the regression analysis of the restricted dependent variables, which used the panel Tobit model, was able

to effectively avoid estimation bias and improve the regression accuracy [67]. The basic form of the panel Tobit model is:

$$y_{it}^* = \beta^T x_{it} + u_i + \varepsilon_{it}$$
$$y_{it} = \begin{cases} y_{it}^*, y_{it}^* > 0 \\ 0, y_{it}^* \le 0 \end{cases} \quad (8)$$

In Equation (8), the following variables are defined. $y_{it}^*$ is the vector of potential dependent variables; $\beta^T$ is the vector of parameters to be estimated; $x_{it}$ is the vector of independent variables; $y_{it}$ is the vector of observed dependent variables; the perturbation term $\varepsilon_{it} \sim N(0, \sigma_\varepsilon^2)$, $u_i$ is the individual effect; $i$ denotes the city; and $t$ denotes the time.

## 4. Results

### 4.1. Spatial and Temporal Evolution of Tourism Industry and urbanization Levels

The above methods were used to measure the tourism industry and urbanization levels in 10 cities in the mountainous border areas of western Yunnan from 2010 to 2019, and their means, standard deviations, and coefficients of variation were calculated (Table 4) in order to measure the overall levels and their regional differences. In addition, the geometric interval method and the natural interruption point grading method were used to grade the tourism industry and urbanization levels in two-time sections in 2010 and 2019 (Figures 3 and 4) in order to explore the evolution of their spatial patterns.

**Table 4.** The related statistics of tourism industry and urbanization levels in the mountainous border areas of western Yunnan.

| Year | Tourism Industry | | | Urbanization | | |
|------|------|------|------|------|------|------|
| | Mean | SD | CV | Mean | SD | CV |
| 2010 | 0.117 | 0.150 | 1.282 | 0.300 | 0.306 | 1.020 |
| 2019 | 0.451 | 0.251 | 0.556 | 0.530 | 0.537 | 1.014 |

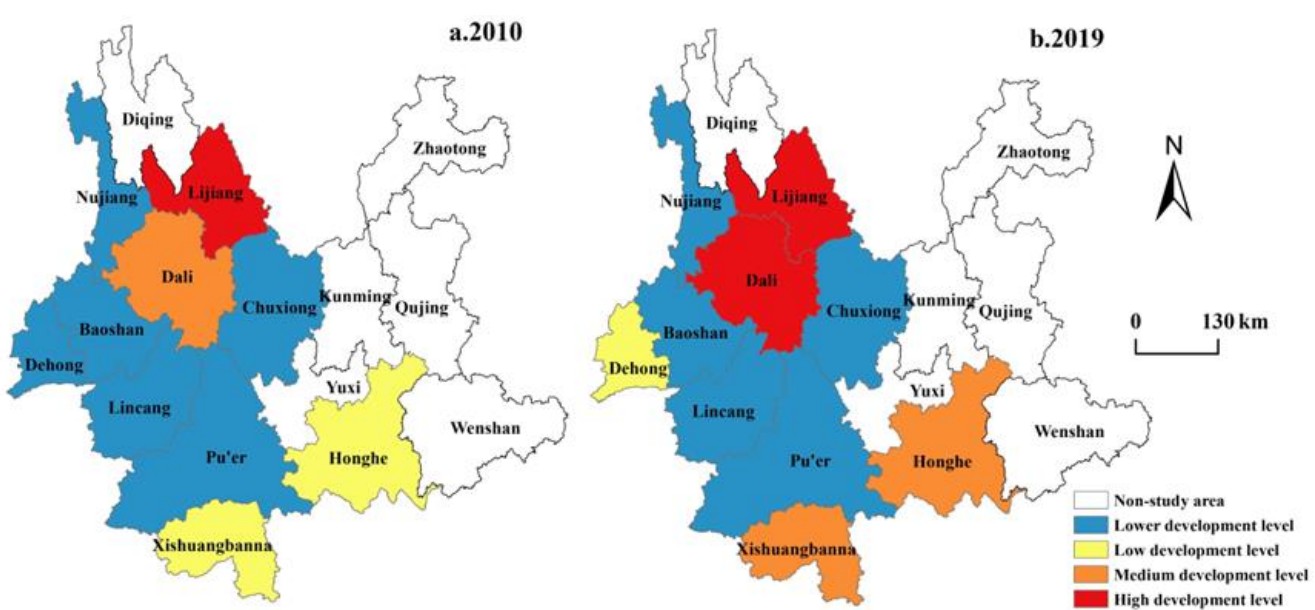

**Figure 3.** The spatial distribution of tourism industry development levels in the mountainous border areas of western Yunnan.

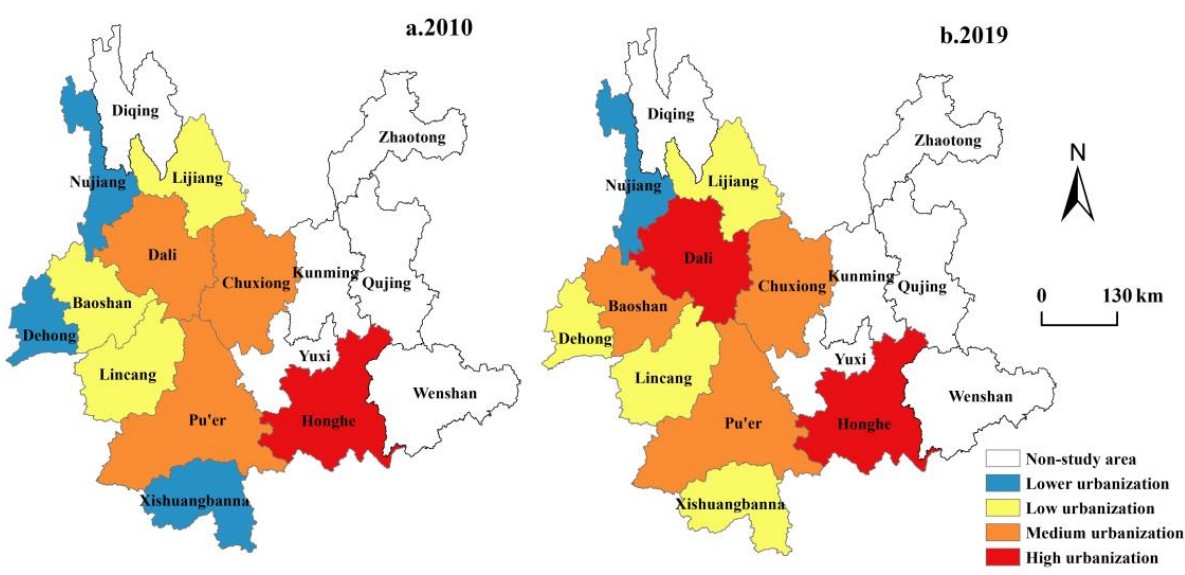

**Figure 4.** The spatial distribution of urbanization levels in the mountainous border areas of western Yunnan.

### 4.1.1. Spatial and Temporal Evolution of Tourism Industry Level

The mean values of the tourism industry level in the mountainous border areas of western Yunnan in 2010 and 2019 were 0.117 and 0.451, respectively, with an overall improvement of nearly three times and an average annual growth rate of 16.2% recorded, which reflect the rapid development of the tourism industry in this region. The standard deviations of the tourism industry level in the mountainous border areas of western Yunnan in 2010 and 2019 were 0.150 and 0.251, respectively, with coefficients of variation of 1.282 and 0.556 obtained, respectively. The absolute gap aspect expanded by nearly 68%, and the relative gap aspect narrowed by nearly 1.3 times. In the study period, as local governments increased their support for tourism development, the overall level of the tourism industry in the study region improved significantly. However, due to the differences in the tourism resource endowment, location conditions and economic foundations of each city, the development base and growth rate of the tourism industry were inconsistent, and the gap in the total scale of the regional tourism industry widened. From Figure 3, it can be seen that: (1) in 2010, there were fewer areas with high tourism development levels, with Lijiang standing out as the leading city with regards to its development of its regional tourism industry. Five of the remaining nine cities were in the low development level stage, of which Nujiang was the most backward region in terms of its tourism industry development level. From an overall perspective, the level of the tourism industry in the entire study region formed a spatial pattern characterized by "prominent poles and low value concentration". (2) In 2019, the number of regions with high tourism industry levels increased to two. Dali City achieved a leap from a "medium level" to a "high level", while Honghe and Xishuangbanna achieved a leap from a "low level" to a "medium level". Although the tourism industry development level of each city had improved, more than half of the areas still had low development levels. At the same time, there was a difference of as much as 15 times between the city with the lowest value, Nujiang, and that with the highest value, Lijiang, and the inherent polarization phenomenon remained prominent. From an overall perspective, the tourism industry level in the entire study region formed a spatial pattern characterized by "high values adjacent to each other, medium values scattered, and low values contiguous".

### 4.1.2. Spatial and Temporal Evolution of Urbanization Level

The mean values of the urbanization level in the mountainous border areas of western Yunnan in 2010 and 2019 were 0.300 and 0.530, respectively, with an overall increase of

nearly two times and an average annual growth rate of 6.54% recorded; the process of urbanization was found to have steadily advanced during the study period. The standard deviations of the urbanization level in the study region in 2010 and 2019 were 0.306 and 0.537, respectively, and the coefficients of variation were 1.020 and 1.014, respectively. The absolute gap aspect expanded by nearly 76%, and the relative gap aspect narrowed by nearly 1%. This indicates that, in the study period, with the continuous promotion of urbanization in the mountainous border areas of western Yunnan, the absolute difference in the urbanization level among the cities was increasing, while the relative difference was shrinking too slowly, making the spatial polarization of urbanization in the region expand and slowing the overall development of the region towards being balanced. From Figure 4, we can see that: (1) in 2010, Honghe was the only region with a high level of urbanization, while three of the remaining nine cities had low levels of development, among which Nujiang, due to its geographical location and resource conditions, had an urbanization level of only 17.67%. On the whole, the urbanization level in the entire study region formed a spatial pattern characterized by a "decreasing step from east to west". (2) In 2019, the number of areas with high levels of urbanization increased to two, with Honghe still in the leading position, and Dali, through relying on the synergistic effect of secondary and tertiary industries, achieving a shift in its urbanization level from a "medium" to a "high" level. Dehong, Baoshan, and Xishuangbanna also achieved a "step" in their urbanization levels, while Nujiang was still at a low urbanization level and was growing more slowly than the other cities. From an overall perspective, the urbanization level in the entire study region formed a spatial pattern characterized by "two prominent poles and fragmented distribution".

*4.2. Spatial and Temporal Evolution of the Coupling Coordination Degree of Tourism Industry and Urbanization*

4.2.1. Temporal Evolution of Coupling Coordination Degree

From Figure 5, it can be seen that the overall tourism industry and urbanization coupling coordination degree in the study region was not high, being between 0.4 and 0.7 during the study period, meaning it was mostly in the primary coordination stage; it therefore still has plenty of room for improvement in the future. From the viewpoint of temporal changes, the coupling coordination degree of the tourism industry and urbanization during the study period showed a good trend of steady growth, and the overall coordination level continued to improve, rising from the primary coordination level (0.405) in 2010 to the intermediate coordination level (0.671) in 2019, with a 65.68% increase in the coupling coordination degree and an average annual growth rate of 5.77% recorded. Alongside the development of the tourism industry and urbanization levels in the study region, the coupling coordination effect of the two tended to be continuously optimized. The level of coupling coordination can be divided into two stages. (1) Firstly, 2010–2016 was characterized by the primary coordination stage, and the coupling coordination degree during this period was low. The reason for this was that the tourism industry and urbanization levels in the study region were not high at this stage, and the interaction between the two was not deeply understood. In addition, the construction of both the tourism industry and urbanization mostly focuses on the pursuit of scale expansion, which is inclined to the rough development mode. (2) Secondly, 2017–2019 was characterized by the intermediate coordination stage, and the degree of coupling and coordination during this period was significantly improved. The reasons for this were mainly the accelerated pace of the construction of a strong tourism industry and urbanization in Yunnan after 2017, the fact that the industrial advantages of the mountainous border areas of western Yunnan as a well known domestic tourist destination began to be highlighted, the construction of the western Yunnan town cluster, which led to a significant improvement in the quality of local urbanization, and the mode of the tourism industry and urbanization integration, which became an important driving force for local economic and social development.

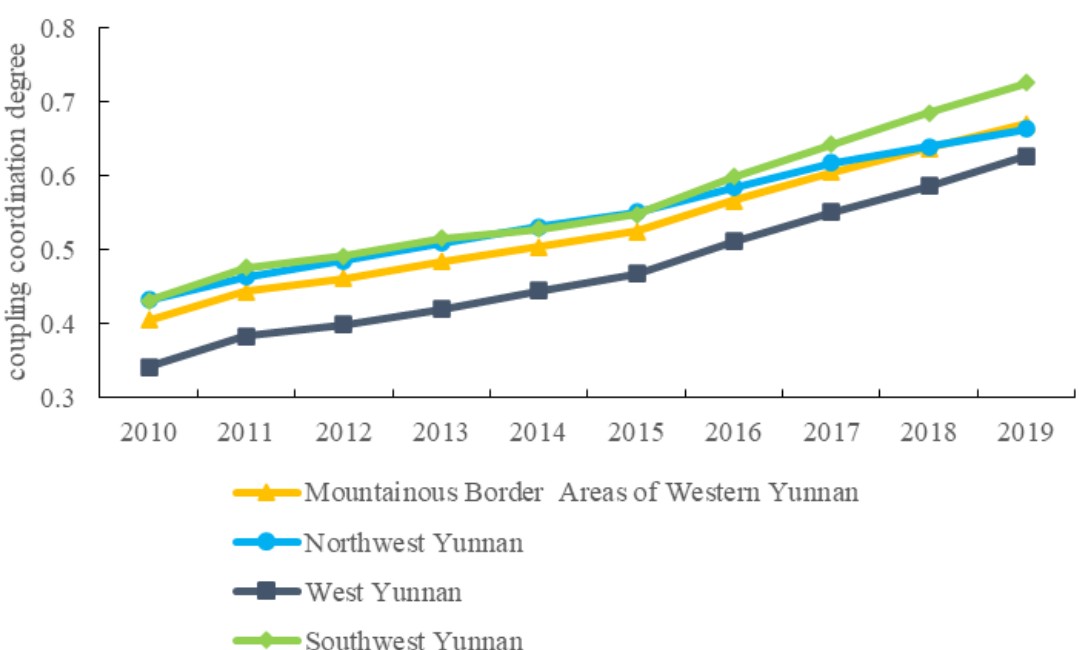

**Figure 5.** The change trend of coupling coordination degree of tourism industry and urbanization.

At the regional level, the coupling coordination degree of the tourism industry and urbanization in the three major regions of the mountainous border areas of western Yunnan increased during the study period [1], but the internal development level was not balanced. This manifested itself as follows. (1) Regarding the means of the coupling coordination degree, the following ranking was obtained: southwest Yunnan (0.564) > northwest Yunnan (0.547) > west Yunnan (0.473), all of which belonged to the minimal coordination level, with there still being plenty of room for future increases in these values. Among the cities, Dali City had the highest mean value of the coupling coordination between its tourism industry and urbanization (0.721), reaching a moderate coordination level, which was due to the fact that Dali City, as one of the first national pilot areas for urbanization, has gradually explored the development path of integration and interaction between its tourism industry and its construction of central cities and characteristic towns. The mean value of the coupling coordination between the tourism industry and urbanization in Nujiang (0.282) was the lowest, being the close to imbalance level, because the development levels of both the tourism industry and urbanization in Nujiang were relatively low, and the benign interaction mechanism between the two systems had not yet been established. (2) In terms of the average annual growth rate of the coupling coordination in the study period, the following ranking was obtained: West Yunnan (7.00%) > Southwest Yunnan (5.95%) > Northwest Yunnan (4.87%). At the same time, the level of the coupling coordination between the tourism industry and urbanization in western Yunnan experienced three stages: close to imbalance (2010–2012); minimal coordination (2013–2018); and moderate coordination (2019). Further, the potential for the interactive development of the tourism industry and urbanization in the region began to be realized due to the role of post-emergence advantages. The level of the coupling coordination between the tourism industry and urbanization in Southwest Yunnan and Northwest Yunnan underwent an evolutionary course of minimal coordination (2010–2016) to moderate coordination (2017–2019) as Dali, Lijiang, Xishuangbanna, and other cities ushered in a period of transformation and upgrading, each focusing more on the quality of their tourism industry and urbanization. Therefore, the growth rate of the coupling coordination level of the tourism industry and urbanization slightly slowed down.

4.2.2. Spatial Evolution of Coupling Coordination

According to the type division shown in Table 3, ArcGIS10.5 software was used to visualize and analyze the coupling coordination degree of the tourism industry and urbanization in the mountainous border areas of western Yunnan. Further, the spatial evolution pattern of the coupling coordination degree from 2010 to 2019 was obtained (see Figure 6). From Figure 6, it can be seen that the coupling coordination degree of the tourism industry and urbanization in the study region had an obvious spatial divergence pattern and a trend of gradual expansion. In 2010, the overall level of the coupling coordination degree was low, and the cities having the minimal coordination level were mainly located in the east, while the cities having the close to imbalance level were mainly located in the west, with this amounting to not only a certain agglomeration phenomenon, but also a decreasing gradient pattern from east to west. At this stage, half of the cities were at the close to imbalance and minimal coordination levels, which was mainly due to the low levels of both the tourism industry and urbanization in these areas, as well as to the fact that they had not yet formed a good interactive development pattern. In 2019, the coupling coordination degree in the mountainous border areas of western Yunnan had greatly improved, with only Nujiang still being at the close to imbalance level. The number of cities in the minimal coordination level had reduced to one, and there were six new cities at the moderate coordination level, as well as two at the good coordination level. However, in terms of spatial patterns, the gap between the cities was still large. Dali and Honghe had gradually become the "poles" with high levels of coupling coordination. However, as the distance between the cities increased, the radiation pulling effect of the two cities on their surrounding areas was relatively weakened, showing a certain "distance–decay effect". At this stage, the proportion of cities at the close to imbalance and minimal coordination levels dropped to 20%, and the coupling coordination between the tourism industry and urbanization in the study region basically reached the ideal state.

According to the subtypes of coupling coordination shown in Table 3, at different levels of coordination, the tourism industry in the study region was mainly obstructed or lagging, which indicates that the development speed of the industry in the region was mismatched, seriously restricting the coordinated and synchronous development of the two systems. In 2010, Nujiang, Baoshan, Dehong, Lincang, and Pu'er were all in the "close to imbalance—tourism industry lagging behind" subtype. At this stage, Nujiang can be taken as a representative city due to the constraints of its location, transportation, and economic level, as well as the fact that its tourism industry started late; therefore, having a poor foundation, the city's coupling and coordination of its tourism industry and urbanization were hindered. Meanwhile, Chuxiong, Honghe, and Dali were in the "minimal coordination—tourism industry lagging behind" subtype. Although the cities represented by Dali were already well known tourist destinations at this stage, there was still some room for its tourism industry to drive its urbanization, while its rapid promotion of urbanization failed to match this. In 2019, Nujiang was in the "close to imbalance—tourism industry lagging behind" subtype. Lincang, Baoshan, Chuxiong, and Honghe were in the "moderate coordination—tourism industry lagging behind" subtype. Lijiang and Xishuangbanna were in the "moderate coordination—urbanization lagging behind" subtype. And Honghe was in the "good coordination—tourism industry lagging behind" subtype. Due to the continuous improvement of infrastructure, the appearance of towns and cities in the mountainous border states of western Yunnan was greatly enhanced over the study period. However, the tourism industry in the region entered a period of transformation and upgrading during this period, and the active role of its tourism industry in the construction of its urbanization needs to be fully realized in the future.

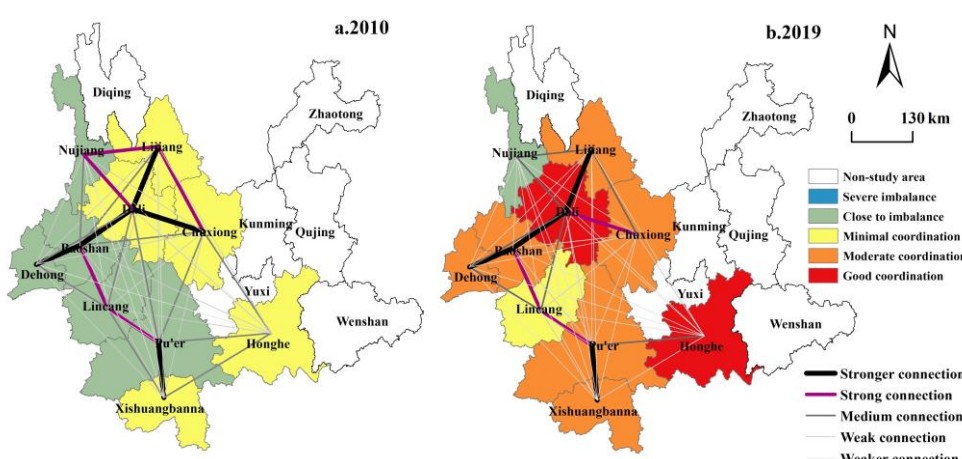

**Figure 6.** The spatial distribution of coupling coordination degree of tourism industry and urbanization.

According to (7), the spatial linkage values of the coupling coordination degree among the cities were calculated and visualized with ArcGIS10.5 software, and they were divided into five levels (see Figure 6). The results show that a more stable spatial network structure of the coupling coordination degree of the tourism industry and urbanization formed among the cities. In 2010, the spatial connection pattern presented two adjacent triangles, with Lijiang–Dali as the common line, Nujiang–Chuxiong as the triangle points, and Lijiang–Dali–Baoshan–Dehong, Pu'er–Xishuangbanna, and Dali–Chuxiong as three main connected axes. In 2019, the spatial linkages had weakened, with a reduction in general linkages and the degradation of some strong and stronger linkages at different levels observed; for instance, the Nujiang–Baoshan, Dali–Pu'er, and Honghe–Xishuangbanna general linkages had been weakened, and the strong linkage of Dali–Chuxiong and the stronger linkages of Nujiang–Lijiang and Nujiang–Dali had been degraded step by step, in turn. In general, the overall spatial connection in the mountainous border areas of western Yunnan was basically ideal, and the two solid axes of Lijiang–Dali–Baoshan–Dehong and Pu'er–Xishuangbanna had been formed. This indicates that there was a certain synergistic effect of tourism industry and urbanization construction in each state and city, producing spatial spillover and radiation drive.

*4.3. Spatial Combination and Evolution Pattern of Coupling Degree and Coupling Coordination Degree of Tourism Industry and Urbanization*

4.3.1. Spatial Combination of Coupling Coordination Types

In order to reveal the differences in the spatial combination of the coupling degree and coupling coordination degree within different time periods, the coupling degrees and coupling coordination degrees of two typical years in 10 cities were classified into spatial combinations (see Figure 7). The following can be seen. (1) From the perspective of time, there were seven types of spatial combinations of coupling coordination in the two typical years: in 2010, there were the severe uncoupling–close to imbalance, slight uncoupling–close to imbalance, slight uncoupling–minimal coordination, advanced coupling–close to imbalance, and advanced coupling–minimal coordination types; in 2019, the advanced coupling–moderate coordination and advanced coupling–good coordination types were added. By comparison, it was found that, in 2010, there were four advanced coupling–minimal coordination types, two advanced coupling–close to imbalance types, and two slight uncoupling–close to imbalance types, while in 2019, there were two advanced coupling–good coordination types and six advanced coupling–moderate coordination type, and the combination type had obviously changed to a high level. (2) From a spatial perspective, the combination types of coupling and coupling coordination changed in each city, and the degree of this change showed some variability. For example, the spatial combination type of Nujiang evolved from the severe uncoupling–close to imbalance type to

the slight uncoupling–close to imbalance type, Lijiang evolved from the advanced coupling–minimal coordination type to the advanced coupling–moderate coordination type, and Pu'er evolved from the slight uncoupling–close to imbalance to the advanced coupling–moderate coordination type, distinctly reflecting the fact that the coupling coordination was continuously strengthening the coupling of the tourism industry and urbanization, leading to a transformation from quantity to quality.

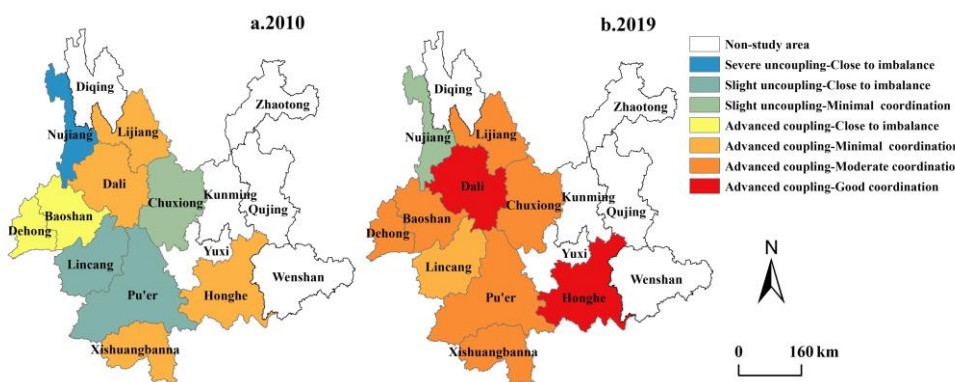

**Figure 7.** The spatial combination of coupling degree and coupling coordination degree of tourism industry and urbanization.

### 4.3.2. Evolutionary Model of Coupling Coordination Types

In order to further summarize the evolution pattern of the spatial combination type of the coupling coordination degree, the evolution pattern of the coupling coordination type of the tourism industry and urbanization in the mountainous border areas of western Yunnan can be summarized as the following. (1) A balanced development pattern. This model involved a spatial combination of the tourism industry and urbanization in the study region, with there being increasing levels and small gaps between them. Further, the spatial combination of the coupling and coordination had a high degree of matching, and the overall development level was more balanced. Only Dali belonged to this model. (2) The inertia development model. This model of regional tourism industry and urbanization development continued to be characterized by the original development inertia, with its systems leading or lagging behind the state and thereby remaining the same. Further, the coupling and coordination of spatial combination in this model were more ideal. Lijiang, Honghe, and Baoshan were typical representatives of this model. (3) The inversion development model. In this model, the tourism industry and urbanization development levels reversed, moving from the original development degree to lagging behind, while its coupling and coordination of spatial combination were general. Dehong and Xishuangbanna were the representatives of this model. (4) The leap development model. The spatial combination of the coupling and coordination of the tourism industry and urbanization in this model was not upgraded step by step, but directly crossed certain stages. Pu'er was the most representative of this model, which directly rose from being on the slight uncoupling-close to imbalance to the advanced coupling-moderate coordination, crossing three stages.

### 4.4. Influencing Factors of the Coupling and Coordination Degree of Tourism Industry and Urbanization

#### 4.4.1. Model Construction and Variable Selection

According to the basic form of the panel Tobit model, and in order to avoid heteroskedasticity and multicollinearity, the explanatory variables were treated logarithmically in order to construct a Tobit model of the factors influencing the coupling and coordination of the tourism industry and urbanization:

$$D_{it} = \beta_0 + \beta_1 \ln is_{it} + \beta_2 \ln ta_{it} + \beta_3 \ln op_{it} + \beta_4 \ln ce_{it} + \beta_5 \ln ca_{it} + \beta_6 \ln ts_{it} + u_i + \varepsilon_{it} \quad (9)$$



In (9), *D* (coupling coordination degree) is the explanatory variable, *β* is the parameter to be estimated for each variable, and *is*, *ta*, *op*, *ce*, *ca*, and *ts* are the explanatory variables. Among them, *is* denotes industrial structure, *ta* denotes transportation accessibility, *op* denotes openness to the outside world, *ce* denotes the capital effect, *ca* denotes consumption capacity, and *ts* denotes talent support, which are characterized by the non-agricultural industries' proportion of the GDP, density of graded road networks, actual utilization of foreign direct investment, per capita fixed asset investment, per capita retail sales of social consumer goods, and number of students in general higher education schools, respectively. The data in this section were obtained from the 2011–2020 Yunnan Statistical Yearbook, and some missing data were filled in by linear interpolation.

### 4.4.2. Regression Model Analysis

In model (1), since the Tobit fixed-effects model cannot account for individual heterogeneity and cannot perform maximum likelihood estimation, a mixed Tobit regression and a random-effects Tobit regression are considered. The likelihood ratio (LR) test rejects the condition of "H0:u = 0" and there is an individual effect, so the random-effects Tobit regression is used (model 1). Tobit regression analysis of the model was performed using Stata software, and the results of the analysis are shown in Table 5.

**Table 5.** The regression results.

| Variate | (1) Random-Effects Tobit Regression | (2) Tobit Regression after Indicator Replacement | (3) Truncated Regression | (4) Random-Effects OLS Regression | (5) Clad Estimate |
|---|---|---|---|---|---|
| lnis | 0.135 (0.043) *** | 0.135 (0.043) *** | 0.195 *** (4.00) | 0.138 *** (3.86) | 0.072 (1.43) |
| lnta | 0.354 (0.088) *** | 0.357 (0.091) *** | 0.264 *** (3.64) | 0.377 *** (5.00) | 0.278 *** (4.30) |
| lnop | −0.025 (0.029) | −0.035 (0.037) | −0.08 (−1.90) | −0.049 ** (−2.03) | −0.087 * (−2.07) |
| lnce | 0.131 (0.058) ** | 0.148 (0.062) ** | 0.223 *** (3.30) | 0.1567 *** (3.35) | 0.342 *** (4.76) |
| lnca | 0.446 (0.091) *** | 0.464 (0.089) *** | 0.352 *** (3.63) | 0.281 *** (3.84) | 0.338 *** (4.26) |
| lnts | 0.115 (0.035) *** | 0.118 (0.036) *** | 0.116 *** (3.43) | 0.097 *** (3.21) | 0.099 ** (2.83) |
| Constant | −0.072 (0.033) ** | −0.088 (0.022) *** | −0.034 (−1.27) | −0.005 (−0.19) | −0.041 (−1.78) |
| sigma_u | 0.03 (0.014) ** | 0.031 (0.0138) ** | | | |
| sigma_e | 0.08 (0.007) *** | 0.079 (0.007) *** | | | |
| LR | *p* = 0.048 | *p* = 0.041 | | | |

Note: models (1) and (2) have standard deviations in parentheses, models (3), (4), and (5) have Z-values in parentheses, and *, **, and *** indicate significance tests were passed at levels of 10%, 5%, and 1%, respectively.

### 4.4.3. Analysis of Regression Results

Using model (1) as the benchmark model for analysis, the results showed the following. (1) Regarding industrial structure (*is*), its regression coefficient was 0.135, and this passed the significance test at the 1% level, indicating that the current industrial structure had a significant impact on the coupled and coordinated development of the tourism industry and urbanization in the study region. The tourism industry is influenced by changes in industrial structure [68], and urbanization also requires the boost provided by the optimization of industrial structure [69]. In the context of the long-term positive economic development in the study region during the study period, the region's industrial structure was gradually upgraded, and the supporting role and thrust role were continuously enhanced. (2) Regarding transportation accessibility (*ta*), its regression coefficient was 0.354, and this passed the significance test at the 1% level, indicating that the improvement of traffic accessibility helped promote the coupled and coordinated development of the tourism industry and urbanization. Transportation conditions provide support for the development of the tourism industry and are an important guarantee for the smooth promotion of urbanization. Due to the accelerated construction of transportation infrastructure in the study region during the study period, the region's relatively occluded transportation shortcomings were gradually alleviated, which helped build the construction of the region's regional tourism industry and urbanization. (3) Regarding openness to the outside world (*op*), its regression coefficient was −0.025, and this did not pass the significance test at the 10% level, indicating that the effect of expanding openness to the outside world on the coordinated development of the tourism industry and urbanization in the study region was not yet obvious. Opening up to the outside world can stimulate the economic vitality

of towns and cities [70] and drive regional development through technology transfer and spillover effects. However, the total amount of foreign investment actually utilized in the study region was extremely low, and it therefore did not play a significant role in the construction of the region's tourism industry and urbanization. (4) Regarding the capital effect (*ce*), its regression coefficient was 0.131, and this passed the significance test at the 1% level, indicating that the strengthening of the capital effect significantly promoted the coupled and coordinated development of the tourism industry and urbanization. With the implementation of policies, such as Western development, the construction of the Belt and Road Initiative, and the policy of promoting the border and enriching the people, the amount of fixed asset investment in the study region grew rapidly over the study period, and the capital involved in its process of tourism industry development and urbanization construction became effectively guaranteed. (5) Regarding consumption ability (*ca*), this was the indicator with the largest influence factor. Its regression coefficient was 0.446, and this passed the significance test at the 1% level, indicating that the improvement of consumption ability was the key influencing factor for the coupled and coordinated development of the tourism industry and urbanization in the study region. The upgrading of residents' consumption is an important means of promoting the development of the tourism industry and an important factor in the improvement of the economic activity of urban markets. In recent years, the living standards of residents in the mountainous border areas of western Yunnan have improved significantly, and their consumption capacity has been enhanced. Furthermore, the demand for tourism and leisure, as well as for other enjoyment and development-oriented areas, has increased, which has injected vitality into the development of the region's tourism industry and urbanization. (6) Regarding talent support (*ts*), its regression coefficient was 0.115, and this passed the significance test at the 1% level. This indicates that talent support played a significant role in the coupled and coordinated development of the study region's tourism industry and urbanization. Improving the quality of the labor force is a factor important to the promotion of long-term economic growth. Due to the vigorous development of scientific education and the introduction of numerous talent introduction policies in the cities in the mountainous border areas of western Yunnan in recent times, the level of these cities' labor force has been solidly improved, providing a sustainable guarantee for the construction of its tourism industry and urbanization.

4.4.4. Robustness Test

In order to verify the accuracy of the analysis results, some of the explanatory variables were replaced with their characterization indicators, and mixed Tobit regression and random effects panel Tobit regression were performed again. The results of the likelihood ratio (LR) test indicated that random effects panel Tobit regression was still required; the analysis results are shown in model (2) in Table 5. Additionally, to further verify the accuracy of the regression results, broken-tail regression (model 3), ordinary panel random effects (using great likelihood estimation, model 4), and clad estimation (model 5) were used for further analysis. By comparing the results, we found little changes in the coefficients and significance levels of the explanatory variables, indicating that the regression results were robust and credible.

## 5. Discussion

### 5.1. Research Findings

In this study, in terms of the development level, the mean values of the tourism industry and urbanization in the mountainous border areas of western Yunnan during the study period were found to have increased by nearly three times and nearly one time, respectively, with the development level having been increased significantly. Further, the standard deviations of the tourism industry and urbanization increased by 68% and 76%, respectively, and the absolute gap in the development levels of the cities was increasing. Further, the coefficients of variation of the tourism industry and urbanization decreased by

nearly 1.3 times and nearly 1%, respectively, and the relative gap in the regional tourism industry was gradually decreasing, while the relative gap in urbanization was slowly shrinking. From the viewpoint of spatial layout, the pattern of the tourism industry development level clearly changed, but more than half of the cities studied were still at the low development level, and the intrinsic two-level differentiation was still obvious. Further, the pattern of the urbanization level in each city also showed obvious improvement, but the phenomenon of spatial polarization expanded, and the overall regional development process moved slowly towards being balanced.

In terms of the sequential variation in the coupling coordination degree, the coupling coordination degree of the tourism industry and urbanization in the border mountainous areas of western Yunnan increased by 65.68% during the study period, showing a good development trend of steady growth. At the regional level, the time series of the coupling coordination degree varied, and the average annual growth rate was ranked as the following: West Yunnan > Southwest Yunnan > Northwest Yunnan. In terms of the spatial evolution of the coupling coordination degree, the location differences in the coupling coordination degree expanded during the study period, evolving from two types to four types; in terms of the subtypes of the coupling coordination degree, the cities maintained consistency, with the tourism industry lagging behind (hindered); in terms of the spatial connection of the coupling coordination degree, two solid axes were formed: "Lijiang–Dali–Baoshan–Dehong" and "Pu'er–Xishuangbanna".

From the spatial combination of the coupling coordination types, each city's combination type levels appeared to be elevated to a certain extent, evolving from five types to seven types. Their spatiotemporal patterns were similar to the differentiation characteristics of the tourism industry development level, urbanization level, coupling degree, and coupling coordination degree. Further, the evolution pattern of the coupling coordination types presented a balanced development pattern, represented by Dali, an inertial development pattern, represented by Lijiang, a reversal development pattern, represented by Xishuangbanna, and a leap development pattern, represented by Pu'er.

From the viewpoint of the influencing factors of the coupling coordination degree, industrial structure, transportation accessibility, the capital effect, consumption capacity, and talent support had significant positive effects on the coupling coordination degree of the tourism industry and urbanization in the mountainous border areas of western Yunnan, with the influence degrees of these factors ranked as the following: consumption capacity > transportation accessibility > capital effect > industrial structure > talent support. Meanwhile, the promotion effect of the degree of opening up to the outside world on the coupling coordination degree was not yet obvious.

*5.2. Practical Implications*

Based on the research results, we propose several recommendations for marginal and less developed regions: First, rely on policies, and find a reasonable orientation based on the background of a given region. By taking advantage of the opportunities brought by the promotion and implementation of regional policies, and by following the principle of "precise positioning, rational exploitation, and development in order", reduce excessive competition and fully explore the development potential of the tourism industry. Further, seize the opportunity provided by the forward position of inland open highlands and practice the urbanization path of scientific development. Second, aim to achieve the effect of "1 + 1 > 2". Guided by the humanistic concept of urbanization and the comprehensive function of tourism, establish a unified mechanism of the "quality improvement" of urbanization and the "transformation and upgrading" of the tourism industry, utilize the interaction and joint effects of the pan-tourism industry, and promote the deep integration and synergistic development of the tourism industry and urbanization. Third, focus on integrated development and the implementation of the "point, line, surface" idea. Take better-developed areas as the essential pivot point, promote the demonstration effect in the process of tourism and urbanization construction, and reduce the cost of

monitoring these areas' vicinities. Further, take the circular tour, town belt, etc., as the key axis, promote infrastructure interconnection, co-construction and sharing, and enhance exchange and cooperation between regions. From point to line, interweave surfaces, make good use of the "key point" and "important lines", establish regional cooperation and coordination mechanisms, implement regional cooperation and development strategies, and promote more balanced overall development in marginal and less developed regions. Fourth, strengthen strengths, fill weaknesses, and optimize the power superimposed effect. Given the reality that some marginal and less developed regions are rich in tourism resources but backward in their economic bases, the development of the tourism industry, through industrial structure upgrading, should be promoted in these regions in order to release economic pressure, promote transit construction in order to enhance traffic accessibility, and improve regional residents' income in order to enhance consumption capacity. At the same time, enhance a given region's financial support ability to ensure the source of funds, improve the quality of foreign investment during the broad opening up stage, vigorously develop science and education, and promote rational personnel policy in order to improve labor quality, thereby deepening the coupling and coordination effect of the tourism industry and urbanization.

*5.3. Further Discussion: The Complex Relationship between the Tourism Industry, Urbanization and the Ecological Environment*

It is necessary to promote urbanization through the development of the tourism industry in marginal and less developed regions, but in this process, the interaction between urbanization and the ecological environment cannot be ignored. On the one hand, the tourism development in marginal and less developed regions mainly occurs in the form of ecotourism, which is dependent on the ecological environment; for example, the mountainous border areas of western Yunnan described in this paper have several national nature reserves, and the existence of these natural resources is the basis for the development of the tourism industry in marginal and less developed regions. On the other hand, urbanization development can negatively affect natural resources and the ecological environment, exacerbate vulnerability, and thus affect the sustainable development of ecotourism. If the urbanization of marginal and less developed regions proceeds into the ecological trap, the chain reaction will be as follows: urbanization destroys the ecological environment, the damage to the ecological environment hinders the development of ecotourism, and ecotourism development affects the urbanization construction, forming a vicious circle, which eventually affects the development of the whole region. Therefore, marginal and less developed regions that are involved in the tourism industry should carefully guide the urbanization process, pay extra attention to the ecological responses, and adopt green routes to urbanization to ensure that urbanization and the development of ecological tourism can interact in a positive fashion.

*5.4. Limitations*

This study did have several limitations. Firstly, we mainly used ArcGIS and the spatial attraction model to analyze the spatial characteristics of the level of tourism industry development, the level of urbanization, and the coupling coordination degree. However, although we attempted to comprehensively reveal the mechanisms behind these phenomena, further analysis is still needed. Multiple methods can be used to supplement and improve relevant research in the future. Secondly, to ensure the study's completeness, the factors influencing the coupling coordination degree were also analyzed. These factors were selected based on the criterion that they significantly impact tourism and urbanization, and we tried to include all critical factors in the study. However, because tourism and urbanization are complex phenomena, there may have been some factors that we did not include and that therefore should be considered in future studies.

## 6. Conclusions

Based on the analysis of the mechanism of the coupling and coordinated development of the tourism industry and urbanization, this study measured the development levels and coupling coordination degree of the tourism industry and urbanization in the mountainous border areas of western Yunnan, as well as their spatial–temporal pattern evolutionary characteristics. Furthermore, using the panel Tobit model, the study explored the influencing factors of the coupling and coordinated development of the tourism industry and urbanization in the study region. The important findings of the study were as follows:

(1)　The development levels of the tourism industry and urbanization in the mountainous border areas of western Yunnan in the study period showed growth trends, but they were uneven in their spatial distributions.

(2)　The coupling coordination degree of the tourism industry and urbanization in the study region also presented a development trend of steady-state growth. However, from the perspective of spatial evolution, the location differences of different cities expanded.

(3)　Combining the coupling degree and coupling coordination degree of different cities, four evolutionary patterns were found: balanced development, inertia development, inversion development, and leapfrog development patterns.

(4)　The influence degrees of the different factors on the coupling coordination degree were ranked as the following: consumption ability > transportation accessibility > capital effect > industrial structure > talent support > opening up to the outside world.

The research results have reference significance for the development of marginal and less developed regions and can provide insights for the promotion of the effective integration development of the tourism industry and urbanization.

**Author Contributions:** Conceptualization, P.Z. and L.Z.; methodology, P.Z. and L.Z.; software, P.Z. and L.Z.; validation, P.Z. and L.Z.; formal analysis, P.Z. and L.Z.; resources, H.Z. and Y.C.; data curation, D.H. and T.W.; writing—original draft preparation, P.Z. and L.Z.; writing—review and editing, L.Z., D.H., and T.W. All authors have read and agreed to the published version of the manuscript.

**Funding:** This research was funded by the National Social Science Fund of China, grant number 20CJY049. The project was also supported by the scientific research fund project of Yunnan Provincial Department of Education (2022Y141).

**Institutional Review Board Statement:** Not applicable.

**Informed Consent Statement:** Informed consent was obtained from all subjects involved in the study.

**Data Availability Statement:** The datasets used in this research are available upon request.

**Acknowledgments:** We thank all the participants involved in the projects for their contributions to our research data.

**Conflicts of Interest:** The authors declare no conflict of interest. The funders had no role in the design of the study; in the collection, analyses, or interpretation of data; in the writing of the manuscript; or in the decision to publish the results.

## Note

[1]　In this paper: the mountainous border areas of western Yunnan were divided into three major regions, northwest Yunnan, west Yunnan, and southwest Yunnan, of which northwest Yunnan includes Chuxiong, Dali, Lijiang, and Nujiang, west Yunnan includes Baoshan, Dehong, and Lincang, and southwest Yunnan includes Pu'er, Xishuangbanna, and Honghe.

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
