# Peer review of "Coupled and Coordinated Development of the Tourism Industry and Urbanization in Marginal and Less Developed Regions—Taking the Mountainous Border Areas of Western Yunnan as a Case Study"

_land, doi:10.3390/land12030640_

Round 1
Reviewer 1 Report
This is an interesting article with a clear and appropriate structure overall. However it could be improved by:
-clarification of language throughout, eg in Section 2 there is mention of 'spiritual enjoyment' (unclear). There are also statements such as 'the operation mode of urbanisation is efficient and scientific' (again unclear), and 'urbanisation is based on social urbanisation (again unclear). What seems to be needed is rigorous proofreading of language and terminology throughout the article (but particularly in section 2) to ensure that there is effective and clear communication to the reader.
-in the research design section there should be clearer explanation and justification of broad research approach (eg use of quantitative methods) as well as the specific methods themselves.
Reviewer 2 Report
Ecological tourism on the territory of protected natural areas is of great importance in the studied area. It is desirable to reflect in the discussion the contradictions and conflicts between the development of ecotourism and urbanization.
Reviewer 3 Report
Reviewer coment:
- This paper correspond for scope of journal. +
- The title corresponds to the content of the paper. +
- This study represents significant contribution for the development of marginal and less developed regions and improving integration development of the tourism industry and urbanization.
-The main question of paper addressed to study coupling and coordinated development of the tourism industry and urbanization, this study measured the development levels and coupling coordination degree of the tourism industry and urbanization in the mountainous border areas of western Yunnan, as well as their spatio-temporal pattern evolutionary characteristics.
- The aim of research is clear, which is noted in abstract. +
- - The aim of research is not pointed out as particular paragraph at the end of chapter of Introduction! The rule is that aim of study need write on the end of chapter of introduction.
- Should be pointed out aim of investigation at the end of Chapter of introduction. *
- Key words are appropriate. +
- Scientific methodology is applied correctly. +
- Results are clearly presented and discussed.
- Tables, figures, pictures are clear.+
- Conclusions are written based on the results.+
- This study represents complementary to the previous ones. +
- Manuscript is acceptable after minor corrections
